# Development of a virtual classroom for pre-analytical phase of laboratory medicine for undergraduate medical students using the Delphi technique

Lena Jafri[1]*, Muhammad Abbas Abid[1], Javeria Rehman[2], Sibtain Ahmed[1], Ghazanfar Abbas[3], Howrah Ali[4], Fatima Kanani[5], Usman Ali[6], Nusrat Alavi[7], Farheen Aslam[8], Sahar Iqbal[9], Aamir Ijaz[10], Muhammad Usman Munir[11], Shabnam Dildar[12], Syed Haider Nawaz[13], Khushbakht Adnan[14], Aysha Habib Khan[1], Adnan Mustafa Zubairi[5], Imran Siddiqui[1]

1 Department of Pathology & Laboratory Medicine, Aga Khan University, Karachi, Pakistan, 2 Department of Education Development, Aga Khan University, Karachi, Pakistan, 3 Shifa International Hospital, Islamabad, Pakistan, 4 Liaquat National Hospital, Karachi, Pakistan, 5 The Indus Hospital, Karachi, Pakistan, 6 Ziauddin Medical University, Karachi, Pakistan, 7 Rahbar Medical & Dental College, Lahore, Pakistan, 8 Quaid-e-Azam Medical College, Bahawalpur, Pakistan, 9 Dow University of Health Sciences, Karachi, Pakistan, 10 Mohi-ud-Din Islamic Medical College, Mirpur, AJK, Pakistan, 11 Armed Forces Institute of Pathology, Rawalpindi, Pakistan, 12 National Institute of Blood Disease & Bone Marrow Transplantation, Karachi, Pakistan, 13 Sindh Institute of Urology & Transplantation, Karachi, Pakistan, 14 Bolan Medical College, Quetta, Pakistan

* lena.jafri@aku.edu

**Data Availability Statement:** Data cannot be shared publicly because of university policy. Data are available from the Aga Khan University Ethics

## Abstract

### Background

Amongst the pre-analytical, analytical, and post-analytical phase of laboratory testing, pre-analytical phase is the most error-prone. Knowledge gaps in understanding of pre-analytical factors are identified in the clinical years amongst undergraduate students due to lack of formal teaching modules on the pre-analytical phase. This study was conducted to seek experts' consensus in Clinical Chemistry on learning objectives and contents using the Delphi technique with an aim to develop an asynchronous virtual classroom for teaching pre-analytical factors of laboratory testing.

### Methods

A mixed method study was conducted at the Aga Khan University. A questionnaire comprising of 16 learning objectives and their associated triggers was developed on Google Docs for developing the case vignettes. A four-point Likert Scale, which included strongly agree, agree, disagree and strongly disagree, was utilized for the learning objectives. An open-ended question was included for experts to suggest new items for inclusion. A cut off of at least 75% agreement was set to establish consensus on each item. A total of 17 Chemical Pathology faculty from 13 institutions across Pakistan were invited to participate in the first round of Delphi. Similar method of response was used in round two to establish consensus on the newly identified items suggested by the faculty in round 1. Later, the agreed-upon

Committee (contact via erc.pakistan@aku.edu) for researchers who meet the criteria for access to confidential data.

**Funding:** The author(s) received no specific funding for this work.

**Competing interests:** The authors have declared that no competing interests exist.

objectives and triggers were used to develop interactive scenarios over Moodle to concurrently test and teach medical students in a nonchalant manner.

## Results

A total of 17 responses were received in Round 1 of the Delphi process (response rate = 100%), while 12 responses were received in Round 2 (response rate = 71%). In round 1, all 16 learning objectives reached the required consensus ($\geq$ 75%) with no additional learning objectives suggested by the experts. Out of 75 triggers in round 1, 61 (81.3%) reached the consensus to be included while 39 were additionally suggested. In 2nd round, 17 out of 39 newly suggested triggers met the desired consensus. 14 triggers did not reach the consensus after two rounds, and were therefore eliminated. The virtual classroom developed using the agreed-upon learning objectives and triggers consisted of 20 items with a total score of 31 marks. The questions included multiple choice questions, fill in the blanks, drag and drop sequences and read-and-answer comprehensions. Specific learning points were included after each item and graphs and pictures were included for a vibrant experience.

## Conclusion

We developed an effective and interactive virtual session with expert consensus on the pre-analytical phase of laboratory testing for undergraduate medical students which can be used for medical technologist, graduate students and fellows in Chemical Pathology.

## Introduction

Quality and safety in diagnostic testing is essential in providing safe and effective patient care [1, 2]. Laboratory medicine plays a vital role in providing accurate results that are crucial to decision making and patient management in clinical practice [3, 4]. The entire total testing process in laboratory practice can be broadly divided in to three phases; the pre-analytical, analytical and the post- analytical phase [1, 2, 5]. According to the ISO 15189: 2012 standard for laboratory accreditation, the pre-analytical phase can be defined as "steps starting in chronological order from the clinician's request and including the examination requisition, preparation of the patient, collection of the primary sample, and transportation to and within the laboratory, and ending when the analytical examination procedure begins" [6]. The analytical phase involves the processes of measurement and analysis of the required analyte, results' technical validation and its release to the laboratory information management systems [5]. Finally, the interpretation, clinical authorization and communication of results to the requesting clinician constitutes the post-analytical phase [5]. Amongst all three, the pre-analytical phase is regarded as the most error prone phase [3, 4, 7]. According to the literature, the analytical errors account for less than 10%, post analytical errors for 18.5 to 47%, whereas the pre-analytical phase accounts for up to 70% of the total laboratory diagnostic errors [1, 3, 4, 7]. Laboratory errors can occur in any step of these phases and can be defined as "any defect from ordering tests to reporting results and appropriately interpreting and reacting on these" [2, 6]. With the advent of technology, automation and improvement in quality assurance and control measures, literature reports a ten-fold reduction in analytical errors in last two decades but pre-analytical issues is still the area of challenge for the laboratory professionals [1, 3, 6, 7]. Such errors may impact the patient care and outcome in-terms of delayed diagnosis, longer

stays at hospital, greater demand on resources and higher cost [3, 5]. Moreover, these errors can lead to more serious hazards of misdiagnosis and mismanagement in clinical practice [5]. The most commonly encountered pre-analytical errors include noncompliance of instructions (like proper fasting, avoiding some special food, adherence of sample collection time, administration of drugs etc.), inaccurate quality and insufficient samples, missing or wrong identification of patient, missing sample or test requisition, inappropriate containers, contamination, inappropriate transport and ineffective storage conditions [2–4, 6].

Despite the increasing attention and importance of prevention of the errors for quality improvement, there is lack of formal education of laboratory medicine in most of the medical school curricula across the globe [8–11]. Studies have reported inadequate knowledge of undergraduate medical students about various phases of pathology testing [9]. Inadequate knowledge in this field, makes medical students more prone to inappropriate test ordering and mistakes in interpreting the results as future health care providers [10]. Lack of formal and sufficient training and knowledge in interns and residents performing the pre-analytical phase, leads to erroneous laboratory testing and results [12]. Moreover, due to insufficient knowledge about pre-analytical variables, the erroneous results if released from the labs may not be recognized as errors by the treating clinicians [11]. Recognition of this gap in education and the mandate from the regulatory bodies necessitate that the students be well equipped with knowledge and trained to identify and control pre-analytical errors.

In educational research, lack of formal curriculum provides the opportunity to use consensus group method to determine the components of a new course development [13]. One of the most widely used consensus method in medical education is the Delphi technique that accounts for approximately 75% studies utilizing consensus group method in this field [13, 14]. The technique was originally developed by Research and Development (RAND) Corporation and is based on "achieving consensual agreement among expert panelists through repeated iterations of anonymized opinions and of proposed compromise statements from the group moderator" [13, 14]. Moreover, the use of internet and convenience of electronic mails has made Delphi method both time- and cost-effective, thereby increasing its use and popularity among researchers [13, 14]. Therefore, this study was conducted to seek experts' consensus in Clinical Chemistry on learning objectives (LOs) and content using the Delphi (or e-Delphi) technique. The aim was to develop an asynchronous virtual classroom using virtual learning environment (VLE) with consensus from subject experts from various institutes for teaching pre-analytical factors of laboratory testing to undergraduate medical students.

## Methods

A mixed method study was conducted at the medical college of the Aga Khan University, Karachi from December 2020 to May 2021. Exemption from ethical approval for this study was obtained from the institutional ethical review committee (2021-5823-15415). The study was conducted in three phases, Phase I was the Delphi technique for curriculum development, phase II consisted of virtual class creation on VLE and in phase 3 the class was conducted followed by feedback from students.

### Phase I: Delphi technique for curriculum development

Two rounds of Delphi were conducted. In the first round, based on the literature search, a questionnaire comprising of 16 learning objectives and their associated triggers for case vignettes was developed on Google Forms. The learning objectives and triggers were extracted after a thorough review of literature on the most common pre-analytical errors in a Chemical Pathology lab [3, 4]. Extracted data was carefully reviewed for relevance to patient safety by the

in-house Chemical Pathology faculty and were included as learning objectives. A total of seventeen Chemical Pathology faculty from thirteen institutions across Pakistan were invited to participate in the study. The panelists were registered Chemical Pathology Fellows with College of Physicians and Surgeons of Pakistan and were involved in active clinical and academic practice associated with a tertiary care facility. Feedback on learning objectives and related triggers was obtained from the selected panel to seek consensus via Google Forms questionnaire through email. A written consent was obtained from all the panel members prior to their participation in the Delphi process. The consent was obtained via email which outlined the Delphi process, expectations from the panel members, and how the data will be used afterwards. A four-point Likert Scale, which included strongly agree, agree, disagree and strongly disagree, was utilized for the learning objectives and the associated triggers were to be selected. An open-ended question was included for experts to suggest new items for inclusion. A cut off of at least 75% agreement was set to establish consensus on each item, in similarity to the use of percentage agreement as reported by literature [15, 16]. Similar method of response was used in round 2 to establish consensus on the newly identified items suggested by the faculty in round 1. The data was only accessible to researchers and the anonymity and confidentiality of the participants was ensured.

## Phase II: Virtual classroom development

Later, the agreed-upon objectives and triggers were used to develop an asynchronous session with interactive case scenarios over VLE (Moodle) to concurrently test and teach year III medical students in a nonchalant manner. Moodle is a user-friendly tool that can develop vibrant classrooms for active learning of students by increasing their participation through visually appealing question stems and explanations [17, 18]. The virtual classroom was developed by the faculty in the Section of Clinical Chemistry, Department of Pathology & Laboratory Medicine at AKU. The faculty received training for the use of Moodle for classroom development and was aided by the university's information technology (IT) department in the process. The virtual classroom was developed using the H5P plugin and consisted of formative quiz of various items built within the content and learning material to stimulate learning and activate prior knowledge. Each case scenario was followed by an interactive quiz or game. The interactive quiz included multiple choice questions, fill in the blanks, drag and drop sequences and read-and-answer comprehensions. Specific learning points to reinforce and clarify the concepts were added after each item and graphs and pictures were included for a vibrant experience.

## Phase III: Virtual classroom conduction/Post activity survey

By definition, the virtual classroom equates to the process that occurs when teacher, learner, problem and knowledge interact through information and communication technologies for the purpose of learning [19]. Each learning objective was made into a clinical case scenario to represent the most common pre-analytical errors pertinent to that case, while all the triggers with that learning objective were included within each case and represented in the case scenarios. The purpose of developing the case scenarios was to create an engaging simulated environment for the students. With the help of learning objectives real or simulated managerial situations were created for the students to identify potential pre-analytical errors and take action. The questions for which clinical scenarios could not be developed were included as a drag-and-drop sequence. Information related to the learning objective and triggers was included in as a slide right after each question so students can review what they just answered. Visual aids, figures, cartoons and mnemonics were included as learning aid to ease the

learning process as compared to the conventional lecture-based classroom. The virtual class was added in a module, Back to Basics, in the beginning of the third year of undergraduate medical; education training. After completion of the virtual class students were directed to a feedback form. Feedback of students was obtained using a questionnaire on VLE, after the completion of session regarding students learning and satisfaction from the session. The students were required to mark their satisfaction on a Likert scale ranging from 1 (not satisfied) to 4 (highly satisfied).

### Statistical analysis

The information was derived from Moodle after the completion of the classroom session. Data was transferred from Microsoft excel to SPSS. For statistical analysis IBM SPSS Statistics version 21 was used. We analyzed binary data and quantitative data separately. Frequencies, means and standard deviations were generated for quantitative data.

## Results

### Phase I: Delphi technique for curriculum development

A total of seventeen responses from the panelist were received in Round 1 of the Delphi process (response rate = 100%), while twelve responses were received in Round 2 (response rate = 71%). The mean work experience of the panelist was found to be 10.8 ± 7.6 SD years.

In round 1, all sixteen learning objectives reached the required consensus (of >75%), with nine LOs (56.25 percent) reaching 100% consensus while the others achieving 82–94% agreement. There were no additional learning objectives suggested by the experts in this round. Out of seventy-five triggers in round 1, sixty-one (81.3 percent) reached the consensus to be included while thirty-nine new triggers were additionally suggested. In 2nd round, seventeen (43.5 percent) of the 39 newly suggested triggers met the desired consensus. Fourteen triggers did not reach the consensus after round one, while 22 triggers did not reach consensus after round two and were therefore eliminated after each round, respectively.

### Phase II: Virtual classroom development

A total of sixteen LOs, with 78 triggers (Table 1) were finalized to develop the virtual session on pre-analytical factors of laboratory testing. The LOs were related to common error prone steps including identification of vacutainer and their properties, correct order of draw, effect of sampling and storage conditions on analytes, effect of certain factors such as temperature, posture, time, food or medicine consumption, etc., on their relevant analytes or procedures, and many more. As shown in Table 1, each LO was further accompanied by the set of most relevant triggers that were agreed upon by the experts and were used to develop the relevant case scenarios and learning material.

### Phase III: Virtual classroom conduction/Post activity survey

Seventy-six students (75.2%) of the class of year III MBBS attended this asynchronous virtual teaching session via Moodle. Feedback was obtained regarding students learning which indicated that 77.6% did not have previous knowledge about pre-analytical phase of laboratory medicine while 93.4% of students learned new information in the session as shown in Figs 1 and 2. Majority of the students (80.4%) were found to be satisfied (57.9%) or highly satisfied (22.4%) from the content and structure of the virtual classroom as shown in Fig 3.

**Table 1. Agreed-upon learning objectives and triggers to be included in the final classroom.**

| Learning Objectives | Triggers for Case Vignettes |
|---|---|
| 1. Identify the vacutainers available in clinical laboratory and their properties | • EDTA<br>• Lithium heparin<br>• Serum separator tubes<br>• Sodium fluoride<br>• Don't take serum calcium in EDTA<br>• Don't take serum lithium in lithium heparin<br>• Lithium Heparin Syringe for Blood Gases<br>• Citrate Tube for Coagulation |
| 2. Identify the correct order of draw for laboratory investigations | • Blood culture tube<br>• Coagulation tube (blue cap)<br>• Serum tube (no additives, red cap)<br>• Heparin tube (green cap)<br>• EDTA tube (lavender cap)<br>• Glycolysis inhibitor (gray cap) |
| 3. Relate the effect of sampling and storage conditions of specimens on analytes like PTH, porphyrins, ammonia, lactic acid | • PTH and temperature<br>• Porphyrin and sunlight<br>• Ammonia and temperature<br>• Lactic acid and temperature<br>• Delay in transport<br>• Shield from light<br>• Tourniquet on for <1min<br>• Blood gases <3 mins arrival |
| 4. Relate the effect of temperature, heparin, improper mixing and air on ABG results | • Air bubble<br>• Venous blood blue<br>• Improper sample mixing<br>• Delay in transport<br>• Ice |
| 5. Apply International Patient Safety Goal One: Identify patients correctly | • Sample switch<br>• 2 patient identifiers<br>• Bar code<br>• Labelling the sample at the bedside labeling |
| 6. Relate time of blood collection, dosage, and other medications interferences on therapeutic drug monitoring | • Time<br>• Dosage<br>• Peak<br>• Trough<br>• Sample with appropriate information of timings (peak or trough)<br>• Chain of custody |
| 7. Apply the 24-hour urine collection protocol | • Complete 24-hour collection of urine<br>• Low urine creatinine<br>• Assure the right timings of collection followed<br>• Discard first sample in morning and collect the first sample next day |
| 8. Relate the effect of posture, temperature, sampling and storage conditions on renin issues | • Lying<br>• Standing<br>• Ice<br>• Delay in transport<br>• Exercise/stress/sitting before sampling |
| 9. Identify the indications for urine amino acids testing | • LPI<br>• Hartnup<br>• Cystinuria<br>• Fanconis Syndrome<br>• RTA<br>• Dirty sample can clog the HPLC column |
| 10. Identify the Triple Test pre-requisites | • Gestational week 14–20 weeks<br>• Ultrasound scan<br>• Number of pregnancies<br>• Age of mother<br>• History of previous positive child/fetus |

(*Continued*)

**Table 1.** (Continued)

| Learning Objectives | Triggers for Case Vignettes |
|---|---|
| 11. Define optimal fasting and relate the effect of food consumption and medications on some analytes like triglycerides, amino acids, Metanephrine, 5-HIAA and VMA | • Triglycerides and fasting<br>• Amino acids and fasting<br>• Metanephrine<br>• 5-HIAA<br>• VMA<br>• Chocolates, coffee, tea, banana |
| 12. Relate Circadian rhythm with hormones | • Cortisol fluctuates<br>• FSH, LH and menstrual cycle<br>• ACTH |
| 13. Relate the effect of prolonged tourniquet on serum potassium | • Relate the effect of prolonged tourniquet on serum potassium, calcium<br>• Effect of prolonged tourniquet on Proteins and albumin |
| 14. Relate the values of serum TSH in patients who are critically ill | • Sick thyroid<br>• Avoid TSH sampling in sick patients<br>• After cardiac event |
| 15. Detect intravenous line contamination using the "IF Glucose > 800 mg/dL AND creatinine < 0.6 mg/dL" rule | • High glucose<br>• High sodium<br>• Detect intravenous line contamination using the "IF Glucose > 800 mg/dL AND creatinine < 0.6 mg/dL" rule<br>• Correlate with tests in hematology |
| 16. Identify spectrophotometric estimate of the level of interference from hemoglobin (hemolysis index), bilirubin (icterus index) and lipids and chylomicrons (lipemia index) as indicators for pre-analytical errors. | • Bilirubin interferes with serum creatinine<br>• Sample processing<br>• High triglycerides can cause falsely low electrolytes |

## Discussion

The topic of 'pre-analytical phase of laboratory medicine' is not a formal component of under-graduate or post-graduate curriculum of medical education in Pakistan. Lack of a formal

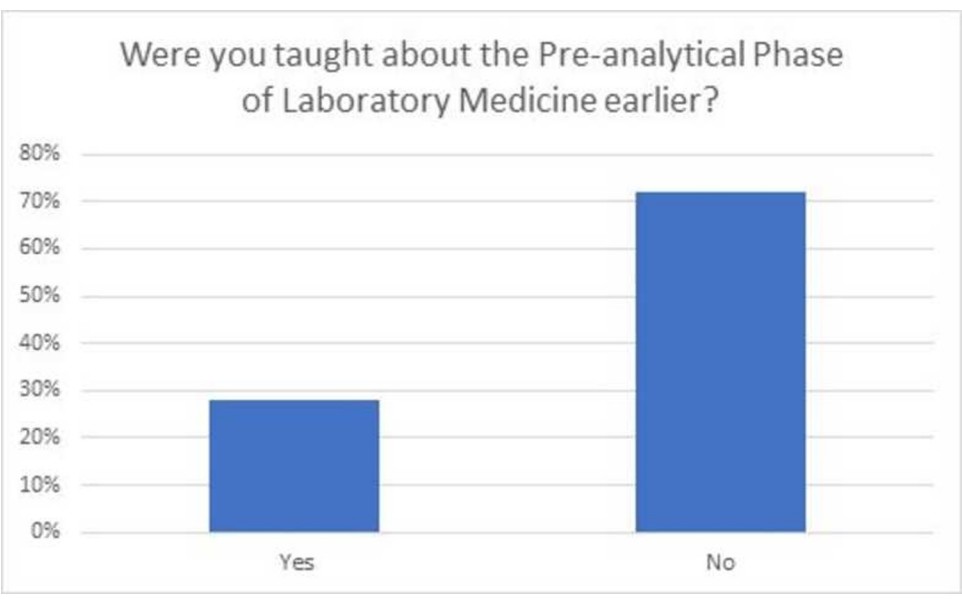

**Fig 1. Students' responses regarding prior knowledge related to the topic.**

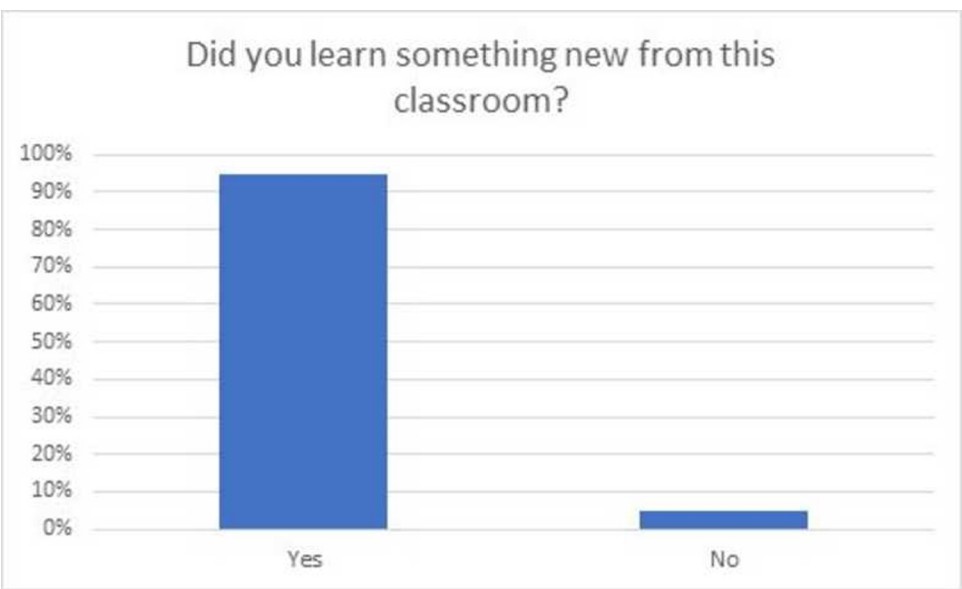

**Fig 2. Survey responses if the class was successful in teaching something new to the students.**

structured modules, gave us the opportunity to seek consensus from various experts on the essential learning objectives for this session. To the best of our knowledge, this is the first study, especially in our local context, to utilize the Delphi technique for developing a formal session on pre-analytical phase and errors of laboratory medicine for undergraduate medical students. The Delphi method was suitable for our study due to its consensus-driven methodology, anonymity, flexibility and cost effectiveness [20, 21]. However, some studies have reported the lack of both standardization in methodology and well-defined reporting criteria as its limitation [13]. Comparable to our study, some studies have used the mixed approaches [15, 22]

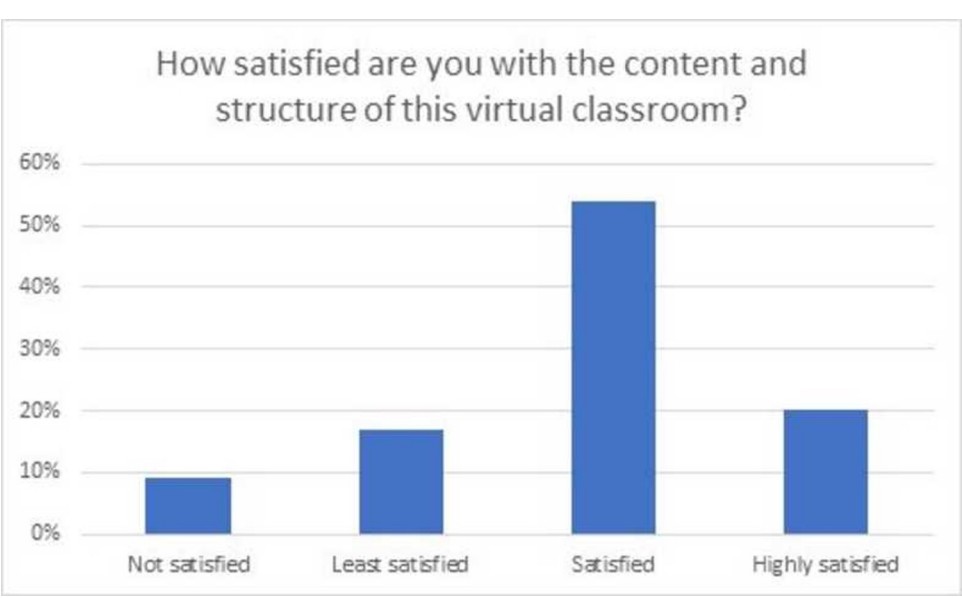

**Fig 3. Survey responses regarding overall satisfaction of students with the developed classroom.**

whereas others have used quantitative or qualitative only, while utilizing Delphi for curricular or training content [23–25].

The LOs and associated triggers identified in our study for developing a session on pre-analytical phase(Table 1) are in agreement with the basic principles of laboratory medicine, as reported by Park & Marques [26]. These LOs and triggers are also in agreement with the most common type of errors of pre-analytical phase of laboratory testing as mentioned by various studies [3, 6, 27, 28]. The virtual session was designed for year III medical students as an interactive, case-based session with a quiz and problem-centered approach to effectively engage and motivate the students. Introduction of such training and information at the level of undergraduate curriculum is supported by studies [29, 30] while some have suggested it in even the pre-clinical years [26, 31] or spread over the entire continuum of curriculum as proposed by Smith et al. [30].

It was interesting to note, that despite no previous formal teaching, 22.4 percent of the students reported to have been taught about pre-analytical phase to some extent. This implies that there may be an element of hidden curriculum or informal teaching which imparts knowledge to the students during pre-clinical years or clerk-ship rotations. The level of satisfaction of majority students in our study reflects utility of our session design in terms of both engaging the students, acquiring new knowledge and fostering their learning. Such level of satisfaction is in agreement with other studies with similar introduction of laboratory medicine teachings [9, 11, 32]. The developed classroom can not only be used for undergraduate medical students, but it can also serve as a useful tool for Pathology & Laboratory Medicine residents, fellows and technologists.

This study also had some limitations. Other experts with lack of internet access and technology limitations may have been missed. Moreover, it was a single session, hence, prior knowledge of students and the gain in knowledge post-session was not determined to assess the effect of the teaching session. Although only clinical pathology specialists may attain a truly comprehensive level of proficiency, it is recommended that all health professional should have the basic understanding of essential principles that guide the choice, interpretation and identification of errors of laboratory tests and results. Hence, virtual session like these can serve the purpose not only in time-effective manner but also for a wide range of learners, ranging from undergraduate, postgraduate students to health professionals at all level of expertise across disciplines as a 'continuing medical education' program. Research is needed to design similar sessions with integration of hands-on experiences and to assess the impact and outcome of such teaching and training with respect to reduction of pre-analytical errors in practice.

## Conclusion

An effective and interactive virtual session with expert consensus on the pre-analytical phase of laboratory testing for undergraduate medical students was developed which can be used for medical technologist, graduate students and fellows in Chemical Pathology and other disciplines.

## Author Contributions

**Conceptualization:** Lena Jafri, Muhammad Abbas Abid.

**Data curation:** Lena Jafri, Muhammad Abbas Abid, Sibtain Ahmed, Ghazanfar Abbas, Fatima Kanani, Usman Ali, Aysha Habib Khan, Imran Siddiqui.

**Formal analysis:** Lena Jafri, Muhammad Abbas Abid, Javeria Rehman, Sibtain Ahmed, Ghazanfar Abbas, Howrah Ali, Nusrat Alavi, Farheen Aslam, Sahar Iqbal, Aamir Ijaz, Muhammad Usman Munir, Shabnam Dildar, Syed Haider Nawaz, Khushbakht Adnan, Aysha Habib Khan, Adnan Mustafa Zubairi, Imran Siddiqui.

**Investigation:** Lena Jafri, Muhammad Abbas Abid, Javeria Rehman, Sibtain Ahmed, Ghazanfar Abbas, Howrah Ali, Fatima Kanani, Usman Ali, Nusrat Alavi, Farheen Aslam, Sahar Iqbal, Aamir Ijaz, Muhammad Usman Munir, Shabnam Dildar, Syed Haider Nawaz, Khushbakht Adnan, Aysha Habib Khan, Adnan Mustafa Zubairi, Imran Siddiqui.

**Methodology:** Lena Jafri, Muhammad Abbas Abid, Javeria Rehman, Sibtain Ahmed, Ghazanfar Abbas, Howrah Ali, Fatima Kanani, Usman Ali, Nusrat Alavi, Farheen Aslam, Sahar Iqbal, Aamir Ijaz, Muhammad Usman Munir, Shabnam Dildar, Syed Haider Nawaz, Khushbakht Adnan, Aysha Habib Khan, Adnan Mustafa Zubairi, Imran Siddiqui.

**Project administration:** Lena Jafri.

**Resources:** Lena Jafri.

**Software:** Lena Jafri.

**Supervision:** Lena Jafri.

**Validation:** Lena Jafri, Muhammad Abbas Abid, Fatima Kanani, Usman Ali, Sahar Iqbal, Aamir Ijaz, Muhammad Usman Munir, Syed Haider Nawaz, Aysha Habib Khan, Imran Siddiqui.

**Visualization:** Lena Jafri, Muhammad Abbas Abid, Ghazanfar Abbas, Howrah Ali, Nusrat Alavi, Farheen Aslam, Sahar Iqbal, Aamir Ijaz, Shabnam Dildar, Syed Haider Nawaz, Khushbakht Adnan, Adnan Mustafa Zubairi.

**Writing – original draft:** Lena Jafri, Muhammad Abbas Abid, Javeria Rehman.

**Writing – review & editing:** Lena Jafri, Muhammad Abbas Abid, Javeria Rehman, Sibtain Ahmed, Ghazanfar Abbas, Howrah Ali, Fatima Kanani, Usman Ali, Nusrat Alavi, Farheen Aslam, Sahar Iqbal, Aamir Ijaz, Muhammad Usman Munir, Shabnam Dildar, Syed Haider Nawaz, Khushbakht Adnan, Aysha Habib Khan, Adnan Mustafa Zubairi, Imran Siddiqui.

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
