## [Decision Letter · Decision Letter 0]

8 Sep 2021

PONE-D-21-24285

Development of a Virtual Classroom for Pre-Analytical Phase of Laboratory Medicine for Undergraduate Medical Students Using the Delphi Technique

PLOS ONE

Dear Dr. Jafri,

Thank you for submitting your manuscript to PLOS ONE. After careful consideration, we feel that it has merit but does not fully meet PLOS ONE’s publication criteria as it currently stands. Therefore, we invite you to submit a revised version of the manuscript that addresses the points raised during the review process.

We look forward to receiving your revised manuscript.

Kind regards,

Gwo-Jen Hwang

Academic Editor

PLOS ONE

Journal Requirements:

a) Did participants provide their written or verbal informed consent to participate in this study?

Reviewers' comments:

Reviewer's Responses to Questions

**Comments to the Author**

1. Is the manuscript technically sound, and do the data support the conclusions?

Reviewer #1: No

Reviewer #2: Yes

2. Has the statistical analysis been performed appropriately and rigorously? 

Reviewer #1: No

Reviewer #2: No

3. Have the authors made all data underlying the findings in their manuscript fully available?

Reviewer #1: No

Reviewer #2: Yes

4. Is the manuscript presented in an intelligible fashion and written in standard English?

Reviewer #1: Yes

Reviewer #2: Yes

5. Review Comments to the Author

Reviewer #1: Dear author,

-methods section, “rounds of Delphi were conducted. In the first round, based on the literature search, a questionnaire comprising of 16 learning objectives and their associated triggers for case vignettes was developed on Google Forms.” But the questionnaire item lack references support and reality and validity, moreover how the development of the study’s related questionnaire is unclear.

-for more contributions, suggest authors add how to Virtual classroom development and are vague, please cite related references.

-a big problem, the study lack a statistic analysis section, to provide the result section.

Reviewer #2: The development of educational content for the pre-analysis phase using Delphi technology, it’s important and meaningful for patient safety. There are some suggestions before publishing.

1. In the first round of Delphi development phase. The authors claim based on the literature search for curriculum development. Finally, a questionnaire comprising of 16 learning objectives and their associated triggers for expert review. Please describe the process of developing the 16 learning objectives and their associated triggers or cite the reference.

2. Of the seventy-five triggers in round 1, 61 (81.3%) reached a consensus for inclusion. However, it is not clear how the triggers that are excluded be treated? Will it be included in the second round of discussion or deleted? As well as how did the 39 new triggers were proposed in 2nd round? In addition to the experts' votes, please describe why they were considered or removed from these triggers.

3. The content of Phase II: Virtual classroom development process is not clearly described. How have the 16 learning objectives and 78 triggers in Table 1 been applied to the learning material in the virtual classroom? How are they different from those of traditional online tests?

---

## [Author Response · Author response to Decision Letter 0]

3 Nov 2021

1. Reviewer #1: Dear author,

-methods section, “rounds of Delphi were conducted. In the first round, based on the literature search, a questionnaire comprising of 16 learning objectives and their associated triggers for case vignettes was developed on Google Forms.” But the questionnaire item lack references support and reality and validity, moreover how the development of the study’s related questionnaire is unclear.

Author’s response: Details about the development of the questionnaire are added in the methods section. Page 4, Line 26: “The learning objectives and triggers were extracted after a thorough review of literature on the most common pre-analytical errors in a Chemical Pathology lab. [15, 16] Extracted data was carefully reviewed for relevance to patient safety by the in-house Chemical Pathology faculty and were included as learning objectives.”

2. -for more contributions, suggest authors add how to Virtual classroom development and are vague, please cite related references.

Author’s response: Details regarding the development of virtual classroom over Moodle have been added to the methods section. Page 5, Line 10: “Moodle is a user-friendly tool that can develop vibrant classrooms for active learning of students by increasing their participation through visually appealing question stems and explanations. [19, 20] The virtual classroom was developed by the faculty in the Section of Clinical Chemistry, Department of Pathology & Laboratory Medicine at AKU. The faculty received training for the use of Moodle for classroom development and was aided by the university’s information technology (IT) department in the process. The virtual classroom was developed using the H5P plugin and, consisted of a formative quiz of various items built within the content and learning material to stimulate learning and activate prior knowledge. Each case scenario was followed by an interactive quiz or game. The interactive quiz included multiple choice questions, fill in the blanks, drag and drop sequences and read-and-answer comprehensions. Specific learning points to reinforce and clarify the concepts were included added after each item and graphs and pictures were included for a vibrant experience.”

3. -a big problem, the study lack a statistic analysis section, to provide the result section.

Author’s response: As suggested, the statistical analysis section has been appropriately added after the methods section. Page 5, Line 3: “The information was derived from Moodle after the completion of the classroom session. Data was transferred from Microsoft excel to SPSS. For statistical analysis IBM SPSS Statistics version 21 was used. We analyzed binary data and quantitative data separately. Frequencies, means and standard deviations were generated for quantitative data.”

4. Reviewer #2: The development of educational content for the pre-analysis phase using Delphi technology, it’s important and meaningful for patient safety. There are some suggestions before publishing. In the first round of Delphi development phase. The authors claim based on the literature search for curriculum development. Finally, a questionnaire comprising of 16 learning objectives and their associated triggers for expert review. Please describe the process of developing the 16 learning objectives and their associated triggers or cite the reference.

Author’s response: Details about the development of the questionnaire are added in the methods section. Page 4, Line 26: “The learning objectives and triggers were extracted after a thorough review of literature on the most common pre-analytical errors in a Chemical Pathology lab. [15, 16] Extracted data was carefully reviewed for relevance to patient safety by the in-house Chemical Pathology faculty and were included as learning objectives.”

5. Of the seventy-five triggers in round 1, 61 (81.3%) reached a consensus for inclusion. However, it is not clear how the triggers that are excluded be treated? Will it be included in the second round of discussion or deleted? As well as how did the 39 new triggers were proposed in 2nd round? In addition to the experts' votes, please describe why they were considered or removed from these triggers.

Author’s response: The proposed changes have been included the results section. Page 6, line 5: “Fourteen triggers did not reach the consensus after round one, while 22 triggers did not reach consensus after round two and were therefore eliminated after each round, respectively.”

6. The content of Phase II: Virtual classroom development process is not clearly described. How have the 16 learning objectives and 78 triggers in Table 1 been applied to the learning material in the virtual classroom? How are they different from those of traditional online tests?

Author’s response: As suggested, the details about the development of the virtual classroom from the agreed-upon learning objectives and triggers have been included in the methods section.

Page 5, Line 25: "Each learning objective was made into a clinical case scenario to represent the most common pre-analytical errors pertinent to that case, while all the triggers with that learning objective were included within each case and represented in the case scenarios. The purpose of developing the case scenarios was to create an engaging simulated environment for the students. With the help of learning objectives real or simulated managerial situations were created for the students to identify potential pre-analytical errors and take action. The questions for which clinical scenarios could not be developed were included as a drag-and-drop sequence. Information related to the learning objective and triggers was included in as a slide right after each question so students can review what they just answered. Visual aids, figures, cartoons and mnemonics were included as learning aid to ease the learning process as compared to the conventional lecture-based classroom.”

---

## [Decision Letter · Decision Letter 1]

17 Jan 2022

PONE-D-21-24285R1Development of a Virtual Classroom for Pre-Analytical Phase of Laboratory Medicine for Undergraduate Medical Students Using the Delphi TechniquePLOS ONE

Dear Dr. Jafri,

Thank you for submitting your manuscript to PLOS ONE. After careful consideration, we feel that it has merit but does not fully meet PLOS ONE’s publication criteria as it currently stands. Therefore, we invite you to submit a revised version of the manuscript that addresses the points raised during the review process.

We look forward to receiving your revised manuscript.

Kind regards,

Vineet Kumar Rai, PhD

Academic Editor

PLOS ONE

Journal Requirements:

Reviewers' comments:

Reviewer's Responses to Questions

**Comments to the Author**

1. If the authors have adequately addressed your comments raised in a previous round of review and you feel that this manuscript is now acceptable for publication, you may indicate that here to bypass the “Comments to the Author” section, enter your conflict of interest statement in the “Confidential to Editor” section, and submit your "Accept" recommendation.

Reviewer #1: (No Response)

Reviewer #2: All comments have been addressed

2. Is the manuscript technically sound, and do the data support the conclusions?

Reviewer #1: (No Response)

Reviewer #2: Partly

3. Has the statistical analysis been performed appropriately and rigorously? 

Reviewer #1: (No Response)

Reviewer #2: N/A

4. Have the authors made all data underlying the findings in their manuscript fully available?

Reviewer #1: (No Response)

Reviewer #2: Yes

5. Is the manuscript presented in an intelligible fashion and written in standard English?

Reviewer #1: (No Response)

Reviewer #2: Yes

6. Review Comments to the Author

Reviewer #1: DEAR Authors,

I suggest that the review’s add my initial opinion with the revised submission an itemized, point-by-point response to the comments of the reviewers.

Thank you.

Reviewer #2: (No Response)

7. PLOS authors have the option to publish the peer review history of their article (what does this mean?). If published, this will include your full peer review and any attached files.

Reviewer #1: No

Reviewer #2: No

---

## [Author Response · Author response to Decision Letter 1]

1 Feb 2022

1/27/2022

Vineet Kumar Rai, PhD

Academic Editor

PLOS ONE 

Subject: Rebuttal Letter-'Response to Reviewers': PONE-D-21-24285R1; Development of a Virtual Classroom for Pre-Analytical Phase of Laboratory Medicine for Undergraduate Medical Students Using the Delphi Technique

Dear Editor,

Please find point by point response to all comments and suggestions made by reviewers.

• Journal Requirements: Please review your reference list to ensure that it is complete and correct. If you have cited papers that have been retracted, please include the rationale for doing so in the manuscript text, or remove these references and replace them with relevant current references. Any changes to the reference list should be mentioned in the rebuttal letter that accompanies your revised manuscript. If you need to cite a retracted article, indicate the article’s retracted status in the References list and also include a citation and full reference for the retraction notice.

Author’s Response: All references have been verified. 

• Reviewers' comments: Reviewer's Responses to Questions

Comments to the Author

1. If the authors have adequately addressed your comments raised in a previous round of review and you feel that this manuscript is now acceptable for publication, you may indicate that here to bypass the “Comments to the Author” section, enter your conflict of interest statement in the “Confidential to Editor” section, and submit your "Accept" recommendation.

Reviewer #1: (No Response)

Reviewer #2: All comments have been addressed

Author’s Response: Not needed

• 2. Is the manuscript technically sound, and do the data support the conclusions?

Reviewer #1: (No Response)

Reviewer #2: Partly

Author’s Response: Not needed

• 3. Has the statistical analysis been performed appropriately and rigorously?

Reviewer #1: (No Response)

Reviewer #2: N/A

Author’s Response: Not needed

• 4. Have the authors made all data underlying the findings in their manuscript fully available?

Reviewer #1: (No Response)

Reviewer #2: Yes

Author’s Response: Not needed

• 5. Is the manuscript presented in an intelligible fashion and written in standard English?

Reviewer #1: (No Response)\\

Reviewer #2: Yes

Author’s Response: Not needed

• 6. Review Comments to the Author

Reviewer #1: DEAR Authors, I suggest that the review’s add my initial opinion with the revised submission an itemized, point-by-point response to the comments of the reviewers. Thank you.

Reviewer #2: (No Response)

Author’s Response to Reviewer One:

Dear Reviewer these were the suggestions and comments shared by you 

“ 1. We note you state consent was obtained from the participants of this study. However, you have not clarified the type of consent provided. Before we can proceed, please address the following prompts:

a) Did participants provide their written or verbal informed consent to participate in this study?

b) If consent was verbal, please explain i) why written consent was not obtained, ii) how you documented participant consent, and iii) whether the ethics committees/IRB approved this consent procedure.”

We have below itemized point-by point response to your comments made then:

Reviewer comment “ 1. We note you state consent was obtained from the participants of this study. However, you have not clarified the type of consent provided. Before we can proceed, please address the following prompts:

a) Did participants provide their written or verbal informed consent to participate in this study?

Author’s Response: Details about the written consent obtained from the participants was included in the manuscript in the methods section. “A written consent was obtained from all the panel members prior to their participation in the Delphi process. The consent was obtained via email which outlined the Delphi process, expectations from the panel members, and how the data will be used afterwards.” (Page 4; Line 35)

b) If consent was verbal, please explain i) why written consent was not obtained, ii) how you documented participant consent, and iii) whether the ethics committees/IRB approved this consent procedure.”

Author’s Response: We did not take verbal consent. Written informed consent was taken from all participants via email and permission from Ethical Review Committee was taken for written consent.

• 7. PLOS authors have the option to publish the peer review history of their article (what does this mean?). If published, this will include your full peer review and any attached files. Do you want your identity to be public for this peer review? For information about this choice, including consent withdrawal, please see our Privacy Policy.

Reviewer #1: No

Reviewer #2: No

Author’s Response: Not needed

Author’s Response: We have uploaded and improved the figure files to the Preflight Analysis and Conversion Engine (PACE) digital diagnostic to ensure that figures meet PLOS requirements

Thank you

Dr Lena Jafri, 

MBBS, FCPS, FAIMER fellow Philadelphia

Department of Pathology and Laboratory Medicine 

Medical College, Pakistan, The Aga Khan University

Stadium Road, P.O. Box 3500, Karachi 74800, Pakistan

---

## [Editor Report · Decision Letter 2]

11 Feb 2022

Development of a Virtual Classroom for Pre-Analytical Phase of Laboratory Medicine for Undergraduate Medical Students Using the Delphi Technique

PONE-D-21-24285R2

Dear Dr. Jafri,

We’re pleased to inform you that your manuscript has been judged scientifically suitable for publication and will be formally accepted for publication once it meets all outstanding technical requirements.

Kind regards,

Vineet Kumar Rai, PhD

Academic Editor

PLOS ONE
---

## [Editor Report · Acceptance letter]

9 Mar 2022

PONE-D-21-24285R2 

Development of a Virtual Classroom for Pre-Analytical Phase of Laboratory Medicine for Undergraduate Medical Students Using the Delphi Technique 

Dear Dr. Jafri:

I'm pleased to inform you that your manuscript has been deemed suitable for publication in PLOS ONE. Congratulations! Your manuscript is now with our production department. 

Kind regards, 

on behalf of

Dr. Vineet Kumar Rai 

Academic Editor

PLOS ONE